# Hospital Care for Cancer Patients—Education and Respect for Patient Rights

**DOI:** 10.3390/healthcare12040494

**Published:** 2024-02-18

**Authors:** Mariola Borowska, Urszula Religioni, Marta Mańczuk

**Affiliations:** 1Department of Cancer Epidemiology and Primary Prevention, Maria Sklodowska-Curie National Research Institute of Oncology, 02-781 Warsaw, Poland; marta.manczuk@nio.gov.pl; 2School of Public Health, Centre of Postgraduate Medical Education of Warsaw, 01-813 Warsaw, Poland

**Keywords:** cancer patient, patient experience, patient rights

## Abstract

This study aims to examine cancer patients’ opinions of safety, the observance of patient’s rights, and the quality of healthcare. Such an analysis will allow for the identification of areas for improvement in quality, safety, and communication between medical staff and patients. Cancer patients are a special kind of patients with chronic and complex diseases, so we need to observe the type of communication they use, which is a critical issue in a hospital ward but also has a significant impact on how the patient follows recommendations at home. Observing a patient’s rights impacts the safety and quality of medical care. This information allows for the identification of areas requiring deeper analysis and improvement. This study was based on a survey conducted at an oncology hospital. The survey contained questions divided into seven sections related to the study areas. Our study emphasizes the importance of knowledge and understanding regarding patient rights among medical staff and patients, underscoring their role in ensuring quality and safety in healthcare settings. We found a strong correlation between the politeness of medical receptionists and staff and patient perceptions of the clarity and exhaustiveness of the information provided.

## 1. Introduction

Hospital care quality and communication regarding cancer patients are critical issues in the treatment of this kind of patient. As cancer is a chronic, complex illness, cancer treatments and communications with cancer patients need to do more than just provide information [1]. Differences in the quality of cancer patients’ care that affect treatment results, patient awareness, and high costs make the study of quality of care increasingly crucial in the context of safety and efficient healthcare [2]. Communication between doctors, nurses, medical technicians in diagnostic labs, other medical staff, the patient, and the patient’s family members is essential to therapy. It determines the quality of care for the patient in the hospital ward and the quality of care provided by their family members at home. Communication between doctors, nurses, medical technicians in diagnostic labs, other medical staff, the patient, and the patient’s family must be about more than just providing information and instruction about preparing for examination or treatment. The traditional model of the doctor–patient relationship is a paternalistic model that limits the autonomy of the patient. This model is part of conventional medicine and is primarily based on the doctor’s responsibility for the patient. Nowadays, a model that places a high value on patient autonomy, the influence of the environment, family relationships, and the patient’s emotional problems related to the illness and beyond is gaining importance. In this model, the doctor interacts with the patient and their family in formulating the diagnosis and planning treatment. It is a system–partner model that precisely emphasizes the role of communication in the therapeutic process. In the system–partner model, there are partnerships between the doctor, the patient, and their family in a relationship that is part of interacting with medical, family, and social systems [1,3,4]. Communication with cancer patients can be based on consultation about examination results, suggested and alternative therapies, talking about feelings and fears about the illness, including the patient in therapy and prognosis, and helping patients and family members find a sense of control and calm [5]. Research about communication between medical staff and patients shows that medical staff with strong communication skills positively influence cancer patients’ satisfaction, improve overall well-being, and affect patients’ experiences. In recent years, research related to the quality of health services has become more and more common. Information from patients has a practical and cognitive aspect. They influence healthcare quality [4,6].

Patient rights are natural rights enjoyed by any patient who requests health services. The history of patient rights is connected to the Universal Declaration of Human Rights, which unequivocally states that every human has rights. This means, among other things, that humans have the inherent right to life, privacy, freedom, free development in society, and respect for dignity [7]. Patient rights are intended to ensure that the patient’s autonomy is protected from interference by others, that adequate conditions are created to exercise these rights, and that those who are obligated to do so comply with these rights [8]. The state guarantees patient rights to everyone. Patient rights are connected with healthcare and describe a manner and standard of performance. Patient rights are, for example, the right to health protection, the right to obtain accessible information about health, the right to immediate medical assistance because of a threat to the patient’s health or life, the right to intimacy and respect for personal dignity during the provision of health services, the right to pastoral care, and others. Patient rights are essential in the medical staff’s professional performance of duties [9,10,11].

Cancer patients’ opinions and experiences of the healthcare system, care, safety, and treatment are increasingly considered essential to informing and improving the quality of care connected with treatment efficacy. Exploring the patient’s experience, listening to their voice, and considering them are important issues in patient-centered care [12]. 

A patient has the right to consent to the provision of healthcare. Consent or lack of consent is possible after obtaining the correct information. The patient can decide by obtaining clear, complete, and understandable information from the medical staff. In a cancer patient’s case, information about treatment, care for the patient after leaving the hospital, and the rules the patient must follow at home are critical in a holistic approach to chronic disease [13]. The information doctors or nurses provide should be in simple language. The doctor should provide information about the diagnosis; suggested and alternative diagnostic and therapeutic methods and the consequences of their use; and the treatment results and prognosis. The patient has the right not to want to know this information. The patient has the right to point to someone authorized to provide information about their health condition [9]. Implementing the patient’s statutory right to information impacts the safety of participation in treatment [10,12].

This study aims to examine cancer patients’ opinions on the observance of patient rights and the quality of healthcare. Such an analysis will allow for the identification of areas for improvement in quality, safety, and communication between medical staff and patients. In a survey, we highlighted the different roles that individual employees, including not only medical staff, play in providing comfort to cancer patients. The role of the medical receptionist is pivotal in shaping the initial experiences of cancer patients. Clear and thorough information about hospital navigation and admission procedures shows a heightened sense of safety and care. The quality of interactions with medical staff, such as doctors and medical technicians, is a crucial component of patient satisfaction. The next critical issue is patient education. For instance, brochures about treatment options and lifestyle advice can be helpful for cancer patients.

## 2. Materials and Methods

This study was based on a survey conducted at an oncology hospital. It was performed in 2021. The sample was purposive. Respondents constituted hospitalized cancer patients who met the inclusion criteria (hospitalization and informed consent to participate in the study). Significantly, the questionnaire noted the need for the patient to evaluate their entire treatment history at this hospital, both their current and previous hospital stay. This is related to the fact that a cancer patient is a patient who returns to the hospital with some frequency. The survey contained 33 closed questions, divided into seven sections, thematically related, and one open-ended question about additional comments, where the patient could write about what was left out of the closed questions. Patients filled out the questionnaire only once and also received an informed consent document. During this process, patients were asked if they had recently filled out the questionnaire, and if so, they were not given it again. Patients were given detailed instructions, requesting that the survey be completed immediately upon receipt and dropped into special boxes marked “QUESTIONNAIRE” placed on the hospital grounds in visible, marked places. The survey was anonymous. Collected answers were used exclusively to compile summary statistics and quality improvement programs. Patients were explicitly asked not to sign the survey with their names.

The first part of the questionnaire was about respecting the patient’s right to information, i.e., whether the medical staff, such as doctors and medical technicians in diagnostic laboratories, informed patients about the possibilities of diagnosis and treatment; whether they involved patients in the treatment process; and whether they spoke clearly and understandably. This study does not have comparative results for the staff who received the highest and lowest ratings regarding the factors under investigation. We did not address this issue because no differences were observed between groups. The second part was about quality issues in care, including whether medical staff treated patients respectfully. The third part dealt with access to psychological care. The fourth part dealt with how patient pain is treated. The fifth part dealt with ancillary services, such as meal delivery and maintaining standards of hygiene and freshness. The quality of the meals was assessed in detail by the patients. Proper food management is critical because diet is an essential part of therapy for cancer patients. The sixth part dealt with cleanliness and the adaptation of wards to the needs of the mobility-impaired and elderly with limited mobility. The seventh part dealt with education and the ability to involve loved ones in the treatment process (e.g., by providing instruction on the care and management of the patient). The last part was an open-ended question. Factors influencing quality and safety in the hospital ward and the patient’s overall opinion of the hospital, according to the patient, were examined.

Responses to the open-ended question were sorted by frequency of occurrence. Answers to the open-ended question were consistent in theme, for example, regarding personnel, room equipment, and work organization on the hospital ward, so they were divided into particular subgroups.

This manuscript describes some results related to the following: -Respecting the patient’s right to information, i.e., whether the medical staff, such as doctors, nurses, and medical technicians in diagnostic laboratories, informed patients about the possibilities of diagnosis and treatment, whether they involved patients in the treatment process, and whether they spoke clearly and understandably;-Quality issues in care, including whether medical staff treated patients respectfully;-Access to psychological care;-Information about how patient pain is treated;-Education and the ability to involve loved ones in the treatment process (e.g., by providing instruction on the care and management of the patient).

The fifth and sixth parts and the last part, an open-ended question, were analyzed in another study. 

This survey analyzed whether medical personnel, such as doctors, nurses, and medical technicians, actively listened to the patient during each conversation and whether information about the patient’s condition, diagnosis, nursing activities, and other factors was simple and understandable. It analyzed the possibility of the patient’s participation in the treatment process. For example, how often during the qualification for various medical procedures and hospitalization doctors informed the patient about possible treatment methods and their consequences.

Answers to individual questions about education and the patient’s right to information were classified on the following scale: definitely yes, mostly yes, mostly no, definitely no, and hard to say. Answers to individual questions about pain management, quality of meals, cleanliness, or ward equipment were classified on the following scale: good, somewhat good, rather bad, definitely bad, hard to say. Positive responses (I strongly agree and rather agree) and negative responses (similarly) were analyzed collectively when formulating conclusions based on the demonstrated relationships.

Statistical analyses were prepared using StatSoft. Inc. (Tulsa, OK, USA) (2017). STATISTICA (data analysis software system) version 13.0. www.statsoft.com (accessed on 20 January 2022). Correlation analysis used Pearson’s chi-squared test and Cramér’s V test to assess the dependence, strength, and direction between variables. All calculations were conducted with a significance level of α  =  0.05.

### Limitations of the Study

Our study has some limitations. One of the limitations is the need for more detailed data about demographic characteristics. Moreover, we focused on hospital care, regardless of the type of hospital ward the patient was staying in, which could affect the interpretation of the study results. In the future, comparing cancer patients’ experiences in inpatient, outpatient, and primary care would be interesting. Our study did not consider nursing care or its evaluation by the respondents. We considered medical receptionists, doctors, and medical technicians. This is a very important consideration that we should take into account in designing our next survey.

## 3. Results

Patients who considered medical receptionists at the institute to be polite also believed that they provided exhaustive and understandable information (*p* < 0.05). Cramér’s V test indicated that this dependence is quite strong.

Figure 1 synthesizes patients’ responses to questions about exhaustive and understandable information provided by medical receptionists and questions about their politeness. For clarity, the figure includes only variants obtained more than ten times. The total number of responses provided was 509.

Patients who positively evaluated the politeness of doctors at the institute (*p* < 0.05), their availability (*p* < 0.05), and respect for patient privacy (*p* < 0.05) also believed that they provided comprehensive and understandable information. Cramér’s V test showed that these dependencies are moderately strong.

Figure 2 is a synthesis of the patients’ responses to questions about the provision of exhaustive and understandable information from doctors and questions about their politeness. For clarity, the chart includes only variants that were obtained more than ten times. The total number of responses provided was 542.

Patients who believed they were adequately informed about the nature of and preparation for radiological examinations agreed with statements that medical receptionists (*p* = 0.00002), doctors (*p* = 0.00012), and medical technicians at the institute (*p* < 0.05) provided clear and comprehensive information. Cramér’s V test indicated that these associations were relatively weak. It was also observed that patients who identified the staff as polite (*p* < 0.05 for medical receptionists and *p* < 0.05 for doctors and medical technicians) considered themselves better informed; however, these associations were also relatively weak.

Patients who positively assessed the politeness (*p* < 0.05) and respect for privacy (*p* < 0.05) of medical technicians at the institute also believed they provided understandable and exhaustive information. Cramér’s V test indicated that the first association is quite strong, while the second is moderately strong.

Figure 3 synthesizes the patients’ responses to questions about exhaustive and understandable information provided by medical technicians and their politeness. For clarity, the figure includes only variants obtained more than ten times. The total number of responses provided was 527. 

In connection with the above, the courtesy of institute staff (regardless of their profession) is critical in conveying understandable information to patients. The courtesy of the medical receptionists relates to the initial contact with the hospital and the provision of information on how to move around the ward. This role is pivotal in shaping the initial experiences of patients. The kindness of the doctors has a powerful influence on the subjective feeling of care and participation in the treatment process. On the other hand, the courtesy of medical technicians, when performing examinations such as CT, determines the feeling of safety during the examination.

Patients who had access to appropriate pain management (*p* = 0.00946) and psychological care (*p* < 0.05) believed that the medical staff responded quickly to their reported ailments. Cramér’s V test showed that the first association is moderately strong, while the second is relatively weak, confirming the particular role of pain management in patient comfort.

Figure 4 is a synthesis of the patients’ responses to questions about the quick response of the medical staff to reported ailments, as well as questions about access to appropriate pain management treatment. For clarity, the figure includes only variants obtained more than ten times. The total number of responses provided was 402. 

Patients who received informational brochures about their illness (*p* < 0.05), tips for living with the disease (*p* < 0.05), and comprehensive information about further treatment (*p* < 0.05) and who had the opportunity to involve close individuals in the treatment process (*p* < 0.05) believed they received satisfactory information about their illness. Cramér’s V test showed that these associations were moderately strong. A weak correlation (*p* = 0.04768) was also found between satisfaction with received information and doctors providing understandable communication about the disease. In summary, attention should be paid to the development of informational brochures about illnesses, informing patients about further treatment plans, and involving close individuals in the treatment process, as these factors play a vital role for patients.

Figure 5 synthesizes the patients’ responses to questions about receiving satisfactory information and informational brochures about their illness. For clarity, the figure includes only variants obtained more than ten times. The total number of responses provided was 413. 

## 4. Discussion

### 4.1. Medical Receptionist Interactions and Initial Patient Experience

The role of the medical receptionist is pivotal in shaping the initial experiences of cancer patients. Our research highlights that patients who encounter polite receptionists who provide clear and thorough information about hospital navigation and admission procedures report a heightened sense of safety and care. This finding aligns with previous studies indicating the importance of first impressions in healthcare settings. The study, involving a total of 3675 patient comments over three years regarding their experiences in undergoing radiologic imaging, shows how important professional staff behavior is. Patient feedback comments were most commonly related to professional staff behavior (74.5%) and wait times (11.9%) and were less widely associated with other issues, such as the quality of the facilities, medical records, cleanliness, and access to information about the exam. This shows how important behavioral elements are. For patients, from the perspective of staff behavior, patient care, courtesy, and professionalism were most important. Hospital staff (medical and non-medical) who were most commonly recognized in the comments were medical technologists (50.2%); receptionists (31.6%); and, much less often, radiologists (2.2%). This shows the role of the medical receptionist is pivotal in shaping the initial experiences of patients. Medical and non-medical staff behavior impacted the patients’ comments about their experiences in undergoing radiologic imaging, especially relating to medical technologists’ and receptionists’ behavior [14].

### 4.2. Doctor–Patient Communication

The quality of interaction with medical staff, particularly doctors, is a crucial component of patient satisfaction. In our study, positive assessments of doctors’ politeness, availability, and respect for privacy correlated with the perception of receiving comprehensive information. This result is consistent with findings from a Croatian study conducted from October 2018 to February 2019 involving 2460 cancer patients. It underscored the need for enhanced communication, especially in conveying sensitive information like bad news. Remarkably, 67% of these patients were unaware of the option to have a family member or friend accompany them during such discussions, suggesting a gap in communication practices [12]. Cancer patients were asked to rate their overall feeling of being cared for on a numeric scale, where very poor care was 0 and excellent care was 10. Almost 38% rated care positively (the numeric scale was noted as 7–10) [12]. Since 2012, the National Cancer Registration Service (NCRS) has collected information on every cancer patient in England. This secure online system is where patients can access the records held by the NCRS on their condition. Cancer patients can talk with doctors about various issues in connection with their illnesses and treatments, for example, information about chemotherapy or other methods of treatment [12]. Moreover, the portal has a quality-of-life tool, a place for more details and notes, helpful information about treatment, and support for chronic disease [15].

### 4.3. Psychological Support and Care

The psychological aspect of cancer care cannot be overstated. Our research indicates that timely access to pain management and psychological support positively impacts patient perceptions of care responsiveness. 

In Croatian research, almost 39% responded that they were informed of whom to ask in all situations when they had concerns about, for example, well-being or self-care. Only 7% of respondents said they had enough care and received other care, such as psychological support [12]. Another study about psycho-oncological care for cancer patients—from the patient’s perspective—identified several barriers connected with psycho-oncological treatment. The most important was low knowledge about psychological support and its role in cancer treatment and poor accessibility [16]. The most popular facilitators were appropriate relationships with medical staff and psychological support. Increasing awareness of psycho-oncological care targeting cancer patients should be one of the most critical goals in treatment [16]. Cancer patients often express a preference for individual over group therapy, pointing to the need for personalized psychological care [17]. Cancer patients frequently suffer psychologically during or after cancer treatment, so psychological care is a crucial step in therapy and self-care. However, barriers such as limited awareness and accessibility to psychological support persist, as evidenced in various studies [18,19,20,21]. Addressing these barriers, including stigma and lack of availability, is crucial for holistic cancer treatment.

A single-center pilot study included family caregivers of advanced cancer patients admitted to the SIPC ward of a university medical center within 12 weeks. Patients were admitted to the SIPC ward because of significant physical and psychosocial symptoms prohibiting further care at home or in non-specialized inpatient wards. Family caregivers of terminally ill cancer patients seem to suffer from relevant psychosocial burdens, including distress and anxiety. These findings indicate the importance of the regular assessment of the psychosocial burden, quality of life, and supportive needs of cancer patient family caregivers. This shows how important the doctor–patient communication model is with regard to taking into account the patient’s family. In this model, the doctor interacts with the patient and their family in formulating the diagnosis and planning treatment. It is a system–partner model that precisely emphasizes the role of communication in the therapeutic process. In the system–partner model, there are partnerships between the doctor, the patient, and their family in a relationship that is part of interacting with medical, family, and social systems. Moreover, psychological care should be provided not only to the cancer patient but also to their families [22].

### 4.4. Patient Education and Involvement

Our findings emphasize the significance of patient education in cancer care. Patients who received detailed brochures about their condition, treatment options, and lifestyle advice and were encouraged to involve their loved ones in the treatment process expressed higher satisfaction with the information provided. They felt adequately taken care of and safe. In therapy, attention should be paid to developing informational brochures about illnesses, informing patients about further treatment plans, and involving close individuals in the treatment process, as these factors play a particularly important role for patients. This finding underscores the need for comprehensive educational materials and family involvement in the treatment process. However, the lack of formal oncology education programs in many European countries, including Croatia, highlights a critical gap in current healthcare practices [12,23,24]. In German research, patients with skin cancer were asked about education provided by medical staff and educational materials about their illness [25]. A prospective study of 461 cancer patients showed the considerable impact of education, which was part of the treatment, on the patients’ quality of life. Additional educational materials on self-care and factors related to living with skin cancer were received by 97 patients. Educational materials allowed them to increase their knowledge and thus awareness of skin cancer factors, such as sun exposure. This research shows that educational brochures are an effective kind in prevention (for skin cancer recurrence) [25]. Another study confirmed that patient education brochures help patients understand their diagnosis and prevent delayed clinical presentation in cases of recurrence [26,27,28]. A scoping review of educational programs about self-care and changing habits in connection with a chronic disease such as cancer showed that education is a critical issue in therapy, for example, educational brochures, psycho-oncological care, teaching methods for dealing with stress, family support, etc. [29].

### 4.5. Implications and Recommendations

These findings have significant implications for healthcare policy and practice. Enhancing initial patient–staff interactions, improving doctor–patient communication, providing comprehensive patient education, and offering personalized psychological support are critical areas for improvement. Healthcare institutions should consider training programs for staff to enhance communication skills, particularly in sensitive contexts. Developing accessible and informative patient education materials and incorporating psychological care into standard cancer treatment protocols are essential steps forward.

## 5. Conclusions

Our study emphasizes the importance of knowledge and understanding regarding patient rights among medical staff and patients, underscoring their role in ensuring quality and safety in healthcare settings.

We found a strong correlation between the politeness of medical receptionists and staff and patient perceptions of the clarity and exhaustiveness of the information provided. This finding underscores the critical role of courteous interactions in effective communication. Patients who felt well informed about radiological examinations and their preparation attributed this to the clear and comprehensive information provided by medical receptionists and clinical staff, including doctors and medical technicians.

The courteous behavior of institute staff, irrespective of their professional role, emerges as a critical factor in conveying understandable and helpful information to patients. This finding highlights the necessity for the systematic training of future medical personnel, especially in communication skills, specifically focusing on interactions with cancer patients.

Additionally, our research highlights the positive impact of providing patients with informational brochures about their illnesses, living tips, and detailed treatment plans. The involvement of patients’ close associates in the treatment process also significantly enhances patient satisfaction and understanding. Therefore, attention should be given to developing and distributing comprehensive educational materials and incorporating family members and friends into the care process.

Regular patient satisfaction surveys are crucial for monitoring adherence to patient rights, assessing service quality, and identifying areas needing improvement. These surveys are instrumental in meeting accreditation standards and ensuring safety in medical institutions.

In conclusion, the findings of our study encourage a holistic approach to patient care that prioritizes effective communication, patient education, and the incorporation of feedback mechanisms to improve the quality and safety of healthcare services continuously.

## Figures and Tables

**Figure 1 healthcare-12-00494-f001:**
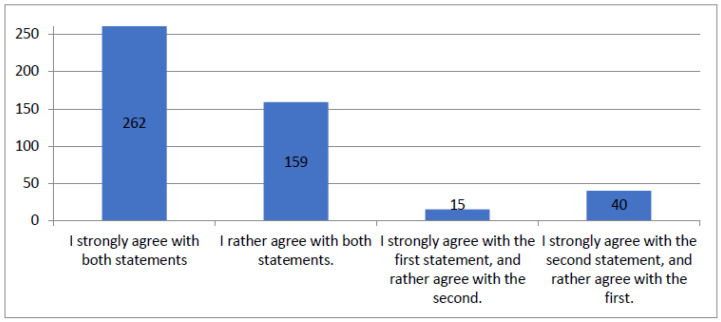
Figure summarizing results of the questionnaire question “Medical receptionists were polite and provided exhaustive and understandable information”.

**Figure 2 healthcare-12-00494-f002:**
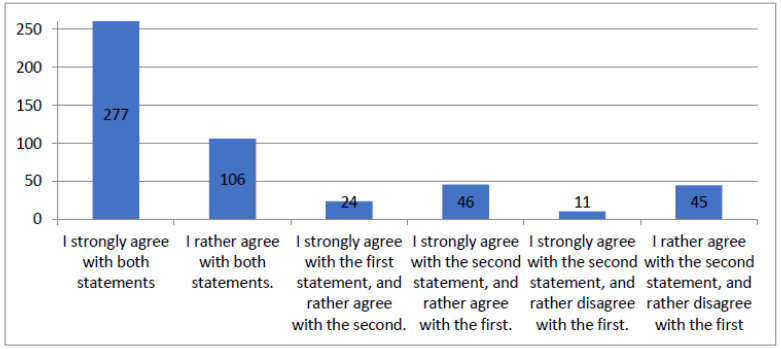
Figure summarizing results of the questionnaire question “Doctors were polite and provided exhaustive and understandable information”.

**Figure 3 healthcare-12-00494-f003:**
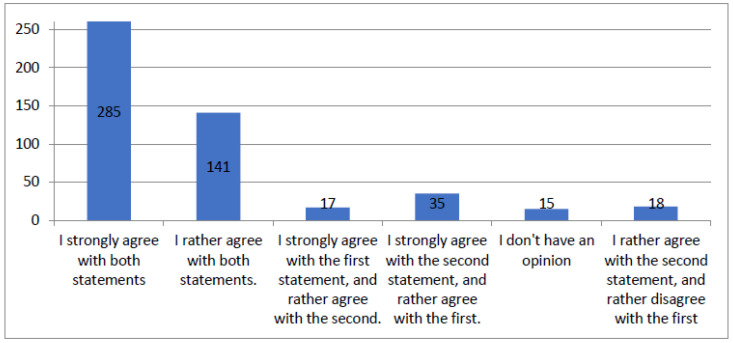
Figure summarizing results of the questionnaire question “Medical technicians were polite and provided exhaustive and understandable information”.

**Figure 4 healthcare-12-00494-f004:**
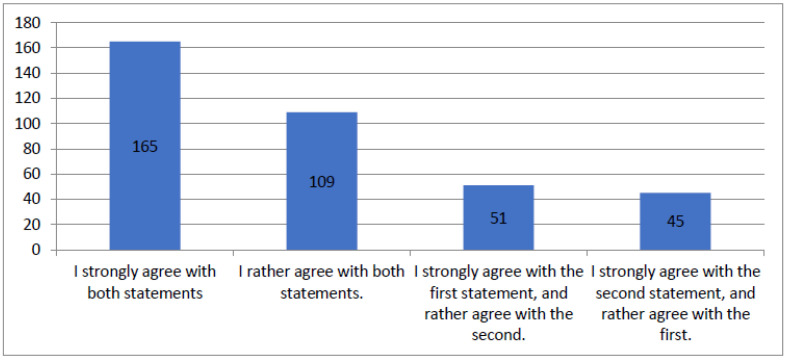
Figure summarizing results of the questionnaire question “Medical staff responded quickly to reported ailments, and I had access to appropriate pain management”.

**Figure 5 healthcare-12-00494-f005:**
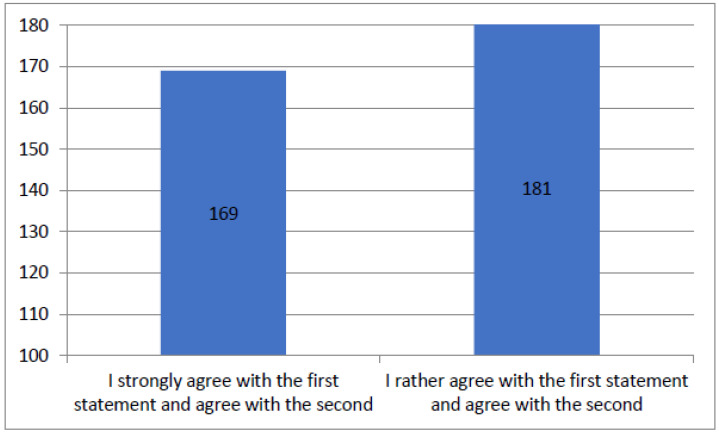
Figure summarizing results of the questionnaire question “I received satisfactory information about my illness and received informational brochures about my illness”.

## Data Availability

The raw data supporting the conclusions of this article will be made available by the authors upon request: mariola.borowska@nio.gov.pl.

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
