# Peer review of "Hospital Care for Cancer Patients—Education and Respect for Patient Rights"

_healthcare, 2024, doi:10.3390/healthcare12040494_

Round 1

Reviewer 1 Report

Comments and Suggestions for Authors

The study aims at analysing cancer patients’ opinions of the safety, observance of rights and quality of heath care. This is an interesting and important question and has the potential to inform hospital policies on patients’ quality of care and communication. The manuscript is well written up to the introduction, after this a series of flaws in clarity, details and reporting is found. The material and methods and results sections require extensive work.

The material and methods section is difficult to follow. A few mistakes, imprecisions and missing information were identified:

-          Line 87 – “The sample was purposive” - what does this mean.

-          Line 88 -  “Respondents constituted hospitalised patients in one of the hospital wards who met the inclusion criteria that included.”  - what were the inclusion criteria? This sentence was interrupted midway.

-          Line 90 – “it’s” – informal.

-          The breakdown of the survey was given twice within the same section. First, between lines 91-93, then in line 99. This should be pulled together and written as one for clarity, synthesis and better flow. Moreover, in line 99 “divided into a few parts” should be more precise – how many parts?

-          Line 116 – “responses to the open-ended questions were sorted by frequency of occurrence" – can this be clarified please.

-          Line 117 – “several types of answers…” – can this sentence be clarified.

-          Line 117 – “this manuscript describes some results related to education and respect for patient’s rights…”If I understand properly this study has been divided into two manuscripts. However, the division into two manuscripts was made only at the result level. If in the introduction and material and methods I am told that a several questions are being tackled, I expect for the results of these questions to be disclosed within the manuscript, and not only a portion of these questions. If the division into two manuscripts is desired then it should be clear from the very beginning that the manuscript under review will be only focusing on a specific aspect of the study and all conclusions are related to that aspect.

Results

-          Can the following sentence be clarified in the context of the results provided: “For clarity, the chart includes only variants that were obtained more than ten times”. What other variants were provided that could have been chosen by the patients? These variants do not match with the ones described in the material and methods sections line 128-132 of the manuscript. Perhaps, because these answers are not considered “individual questions”? If that what the case can the options provided for these questions be outlined as well.

-          For all the p=0.00 can the actual p values be provided.

-          Can figures be referred to as figures rather than charts?

-          Can all figures have a summary of the actual numbers in each bar?

-       How was the correlation conducted? How was the data summarised for the correlation analysis given the many categories/variables. To clarify, were the strongly agree and agree on both statement summarised as one?

The discussion is well written, but I am not sure if it can be backed up by the results obtained as these were not reported fully and clearly.  

Comments on the Quality of English Language

The introduction is well written and exhaustive. The mention to the human rights could be abbreviated and connected to patients’ rights slightly better. There are a few weirdly structured sentences that could be improved e.g. lines 35-36 e.g. more than just providing information and instruction about preparing to examine, line 38 participating patient in therapy.

The material and methods, and results section would highly benefit from re-writing for clarity.

Although the quality of English is sufficient for the majority of the manuscript to be understandable, there are a few badly structured sentences, unfinished sentences and wrong use of certain words impacting the clarity of the manuscript. 

Author Response

Reviewer 1

The study aims at analysing cancer patients’ opinions of the safety, observance of rights and quality of heath care. This is an interesting and important question and has the potential to inform hospital policies on patients’ quality of care and communication. The manuscript is well written up to the introduction, after this a series of flaws in clarity, details and reporting is found. The material and methods and results sections require extensive work. - Thank you very much for your comment.

The material and methods section is difficult to follow. A few mistakes, imprecisions and missing information were identified:

-          Line 87 – “The sample was purposive” - what does this mean. Thank you very much this comment. Our point was that the trial involved patients of a single cancer hospital.

-          Line 88 -  “Respondents constituted hospitalised patients in one of the hospital wards who met the inclusion criteria that included.”  - what were the inclusion criteria? This sentence was interrupted midway. Thank you very much for this question. The inclusion criteria was hospitalization and consent to study. We completed in line 98-100.

-          Line 90 – “it’s” – informal. We sincerely apologize for this wording. We have corrected it.

-          The breakdown of the survey was given twice within the same section. First, between lines 91-93, then in line 99. This should be pulled together and written as one for clarity, synthesis and better flow. Moreover, in line 99 “divided into a few parts” should be more precise – how many parts? Thank you very much for your comment. We have corrected and explain about how many part the questionnaire was divided. Line 103-104.

-          Line 116 – “responses to the open-ended questions were sorted by frequency of occurrence" – can this be clarified please. An analysis of the answers to the open-ended question can be found in the second manuscript currently in preparation. Frequency of occurrence means that we noticed some groups of answers to the open-ended question, consistent in theme, for example, regarding personnel, room equipment or work organization on the hospital ward. Line 128-130.

-          Line 117 – “several types of answers…” – can this sentence be clarified. An analysis of the answers to the open-ended question can be found in the second manuscript currently in preparation. Frequency of occurrence means that we noticed some groups of answers to the open-ended question, consistent in theme, for example, regarding personnel, room equipment or work organization on the hospital ward. Line 128-130.

-          Line 117 – “this manuscript describes some results related to education and respect for patient’s rights…”If I understand properly this study has been divided into two manuscripts. However, the division into two manuscripts was made only at the result level. If in the introduction and material and methods I am told that a several questions are being tackled, I expect for the results of these questions to be disclosed within the manuscript, and not only a portion of these questions. If the division into two manuscripts is desired then it should be clear from the very beginning that the manuscript under review will be only focusing on a specific aspect of the study and all conclusions are related to that aspect. Thank you very much for your comment. Yes, our study has been divided into two manuscripts. We have indicated in other elements of the manuscript that it deals with selected elements of the study: respect for patient’s rights, quality of care, access to psychological care and patient education including access to educational brochures. This manuscript is about the elements in sections 1,2,3,4 and 7. The remaining sections, that is, 5,6 and the open question, are described in the second manuscript, which we are currently preparing. The fifth part dealt with ancillary services, such as meal delivery and maintaining standards of hygiene and freshness. The quality of the meals was assessed in detail by the patients. Proper food management is critical because diet is an essential part of therapy for cancer patients. The sixth part dealt with cleanliness; local conditions were also asked about the adaptation of wards to the needs of the mobility-impaired and elderly with limited mobility. The last part was an open question. The introduction is prepared in the context of only those elements of the study that are presented in the results, that is, the part of the study that is analyzed in this manuscript. We do not write in the introduction about the quality of the meals, or about the labeling of the wards and cleanliness, as well as adaptation to the needs of people with disabilities. Line 98-100; 128-144.

We have also corrected the abstract. Line 18.

Results

-          Can the following sentence be clarified in the context of the results provided: “For clarity, the chart includes only variants that were obtained more than ten times”. What other variants were provided that could have been chosen by the patients? These variants do not match with the ones described in the material and methods sections line 128-132 of the manuscript. Perhaps, because these answers are not considered “individual questions”? If that what the case can the options provided for these questions be outlined as well.  Thank you very much for this suggestion. All charts were made based on the values from a multi-way table combining the answers to the two questions given in the title of the chart. Each question had 5 answers (from strongly agreeing with the statement to strongly disagreeing with it), so there were (hypothetically) 25 possible variants (e.g. strongly agreeing with the first and second statement, strongly agreeing with the first but partially agreeing with the second statement, etc.). Placing all the variants on the chart would make it unreadable, so we only included the variants that appeared more often - as you can see in each chart, only a few variants appeared more than 10 times, two of them appeared much more often than others. Line 170-176; 191-202; 209-222; 230-239.

-          For all the p=0.00 can the actual p values be provided. Thank you for this suggestion. In most cases the "p" value was very low - the first significant number appeared only at a relatively far place after the decimal point, hence we used this notation to increase transparency. However, of course, as suggested, the "p" values have been changed to exact values.

-          Can figures be referred to as figures rather than charts? Thank you very much for your comment. We have corrected.

-          Can all figures have a summary of the actual numbers in each bar? Thank you very much for your comment. We have corrected.

-       How was the correlation conducted? How was the data summarised for the correlation analysis given the many categories/variables. To clarify, were the strongly agree and agree on both statement summarised as one? In order to examine correlations, a multi-way table was prepared, the results of which were performed using the Pearson's Chi^2 test, and if a correlation was found, the V. Cramer's test was performed. Positive responses (I strongly agree and rather agree) and negative responses (similarly) were analyzed collectively when formulating conclusions based on the demonstrated relationships.

The discussion is well written, but I am not sure if it can be backed up by the results obtained as these were not reported fully and clearly. Thank you very much for your comment. We have corrected results. Line 170 – 239. Also we add examples in discussion. Line 253-266; 306-320.

The introduction is well written and exhaustive. The mention to the human rights could be abbreviated and connected to patients’ rights slightly better. There are a few weirdly structured sentences that could be improved e.g. lines 35-36 e.g. more than just providing information and instruction about preparing to examine, line 38 participating patient in therapy. Thank you very much for your comment. We have completed what we meant in this sentence. Line 35-37. Here we have described what the communication of medical personnel with the patient can consist of - among other things, to make the patient feel, through clear, understandable and emphatic communication, that he is participating in the treatment process, that he is an important part of it. This refers to the system-partner model in doctor-patient communication. Nowadays, the paternalistic model is being abandoned in favor of precisely the system-partner model. Line 38 – 48.

The material and methods, and results section would highly benefit from re-writing for clarity. Thank you very much for your comment. We have corrected material and methods and results.

Although the quality of English is sufficient for the majority of the manuscript to be understandable, there are a few badly structured sentences, unfinished sentences and wrong use of certain words impacting the clarity of the manuscript. Thank you very much for your comment. We have corrected results. We have done a correction of English in all manuscript.

Reviewer 2 Report

Comments and Suggestions for Authors

Thank you for the opportunity to review the article "Hospital Care for Cancer Patients – Education and Respect for Patient’s Rights." Regular patient satisfaction surveys are crucial for monitoring adherence to patient rights, assessing service quality, and identifying areas needing improvement. My suggestions for supplementing the article include providing information on: 1. The specific department the study pertained to – whether it was a procedural or non-procedural department; 2. Was ethical committee approval sought, and was permission obtained to conduct the study?" 3. Clarifying whether patients filled out the survey only once or had the option to complete it multiple times, and how this aspect was verified; 4. Specifying whether the study considered nursing care and its evaluation by respondents; 5. Noting the absence of a comparative result for the staff regarding the factors under investigation – who received the highest and lowest ratings

Author Response

Reviewer 2

Thank you for the opportunity to review the article "Hospital Care for Cancer Patients – Education and Respect for Patient’s Rights." Regular patient satisfaction surveys are crucial for monitoring adherence to patient rights, assessing service quality, and identifying areas needing improvement. My suggestions for supplementing the article include providing information on:

  1. The specific department the study pertained to – whether it was a procedural or non-procedural department; Thank you very much for your comment. The study was carried out in a cancer hospital, for all patients, without division into wards.
  2. Was ethical committee approval sought, and was permission obtained to conduct the study?" Thank you very much for your comment. We confirm that the work and conducted research are in accordance with the Declaration of Hel-sinki. Informed consent was obtained from all subjects involved in the study. The Bioethics Committee operates in accordance with the Law on the Medical Profession - the Law of December 5, 1996 on the Profession of Physician and Dentist. Article 29 Opinion of the Bioethics Commission on a medical experiment project. In this article there is information about the necessity of obtaining the opinion of the Bioethics Commission for medical experiments. Our study does not meet the definition of a medical experiment and therefore I do not have the opportunity to obtain the commission's approval, because there is no legal basis for this.
  3. Clarifying whether patients filled out the survey only once or had the option to complete it multiple times, and how this aspect was verified; Thank you very much for your comment. Patients filled out the questionnaire only once, also receiving an informed consent document. During this process, patients were asked if they had recently filled out the questionnaire, and if so, they were not given it again. Patients were given detailed instructions, requesting that the survey be completed immediately upon receipt and dropped into special boxes marked "QUESTIONNAIRE" placed on the hospital grounds in visible, marked places.
  4. Specifying whether the study considered nursing care and its evaluation by respondents; Thank you very much for your comment. Our study not considered nursing care and its evaluation by respondents. We considered medical receptionist, doctors and medical technicians. This is a very important consideration that we should take into account in designing our next survey.
  5. Noting the absence of a comparative result for the staff regarding the factors under investigation – who received the highest and lowest ratings. Thank you very much for your comment. We did not address this issue because no differences were observed between groups.

Round 2

Reviewer 1 Report

Comments and Suggestions for Authors

The introduction is clear and informative. At the moment, the introduction has several repetitive sections/sentences. Work should be done to improve flow and render the introduction more concisive.

Examples:

Line 38/39 – A bit repetitive. Could be more to the point.

Line 43/44 – Would advise the use of their instead of his/hers. The same in line 47.

Lines 81/82 – Repetitive could be summarised into a single sentence.

Last paragraph and beginning of introduction – Repetitive. The final paragraph of the introduction could focus on introducing the study rather than repeating the concepts introduced at the beginning of the introduction.

Material and Methods

Line 98-99 – “Respondents constituted hospitalized patients 98 in one hospital wards who met the inclusion criteria including hospitalization and in- 99 formed consent to participation in the study.” – very confusing sentence – Should be better rephrased.

Line 100-101 – By the questionnaire marked do you intend that the questionnaire required patients to take into account their entire treatment history at the hospital when answering/evaluating?

Line 108 – A few verbs referring to actions in the present are used in their future form – this should be amended e.g. Collected answers will only be used to compile summary statistics and quality improvement programs.

Line 131 – Some is very vague. Highly discouraged from using some. If the results analysed were from section 1-4 of the questionnaire that should be said.

Line 145-146 – 145 “Realization of the patient’s rights to information results from Article 9 of the Law on 145 Patient Rights and Patient Ombudsman.” – sentence not related to the material and methods.

“Positive responses (I strongly agree and rather agree) and negative responses (similarly) were analyzed collectively when formulating conclusions based on the demonstrated relationships”

This explanation of how the answers to questionnaire were summarised should be included in the material and methods as part of the explanation on how the results were analysed.

Results

For all the p=0.00 can the actual p values be provided. Thank you for this suggestion. In most cases the "p" value was very low - the first significant number appeared only at a relatively far place after the decimal point, hence we used this notation to increase transparency. However, of course, as suggested, the "p" values have been changed to exact values. – Thank you for the clarification if the number is too small to be fully reported, then p<0.05 should be provided as standard of reporting rather than p=0.00 or p=0.00000).

The figures are still referred to as charts.

The figure legends should be informative of what the figures are showing and not a statement interpreting the data. E.g. Figure summarising results of the questionnaire question “I received satisfactory information about my illness and received informational brochures 245 about my illness.”

Comments on the Quality of English Language

Please read through typos and missing words. E.g Patient feedback comments are most commonly related to professional staff behavior (74.5%) 256 and wait times (11.9%). –

Author Response

Thank You very much for all your suggestions. We are very happy to make all the corrections. Details we attached in the cover letter. 
